# In Vitro Rooting of *Capparis spinosa* L. as Affected by Genotype and by the Proliferation Method Adopted During the Multiplication Phase

**DOI:** 10.3390/plants9030398

**Published:** 2020-03-23

**Authors:** Valeria Gianguzzi, Ettore Barone, Francesco Sottile

**Affiliations:** 1Department of Agricultural, Food and Forest Sciences (SAAF), University of Palermo, 90128 Palermo, Italy; valeria.gianguzzi@unipa.it (V.G.); ettore.barone@unipa.it (E.B.); 2Department of Architecture (DARCH), University of Palermo, 90128 Palermo, Italy

**Keywords:** caper, biotypes, micropropagation, in vitro rooting

## Abstract

The in vitro rooting of three caper (*Capparis spinosa* L.) selected biotypes, grown in a commercial orchard on the Sicilian island of Salina (38°33′49” N), was performed using—as base material for rooting experiments—shoot explants proceeding from two different in vitro culture systems: solid medium and liquid culture in a PlantForm bioreactor (TIS). The regenerated shoots of each accession were submitted to different auxin treatments (NAA, IBA, IAA - 1 or 2 mg L^−1^; NAA+IBA 0.75 and 0.25 mg L^−1^, respectively), supplemented with sucrose or fructose (mg L^−1^). The highest rooting rate in terms of root percentage (67%) was reached with the explants of the selected accession ‘Sal 39’ proceeding from liquid culture in PlantForm and induced in the MS medium with sucrose, as a carbon source, supplemented with NAA 0.75 mg L^−1^ + IBA 0.25 mg L^−1^, after six days in a climatic growth chamber at 25 ± 1 °C in the dark and then placed under a cool white fluorescent lamp, with a PPFD of 35 μmol m^−1^ s^−1^ and a photoperiod of 16 h. On the other hand, poor rooting rate was generally achieved under all the tested experimental conditions with the other biotypes, ‘Sal 37’ and ‘Sal 35’, demonstrating the strong role exerted by the previously adopted proliferation method and by the genotype for successful caper in vitro rooting.

## 1. Introduction

The caper (*Capparis spinosa* L. Capparidaceae) is an endangered species, at risk of genetic erosion, included among the neglected and underutilized plant species (NUS) [1] but well adapted to the adverse environment of the Mediterranean summer [2]. In this region, this perennial xerophytic shrub is widely diffused both in the wild and the cultivated form and, if properly promoted, could represent a significant contribution to the development of local areas [3]. It may also be helpful to preserve the equilibrium of semiarid lands’ fragile ecosystems [4] by reducing soil erosion hazard [5] and by preventing fires from spreading [6]. In Italy, caper cultivation is mainly concentrated in the small islands around Sicily (mostly Pantelleria and the Aeolian Islands) where the species plays an important economic role [7].

In order to improve caper yield and quality—and, therefore, farmers’ return—propagation systems play a role of paramount importance. Nowadays, vegetative methods are generally preferred over the seed usage for the propagation of selected caper biotypes due to the high degree of seed heterozygosity [8] and the poor seed germination rate. Nevertheless, caper still remains a difficult-to-root woody species [9] where the percentage of conventional rooting rarely exceeds 50% [10]. The results of vegetative propagation by stem cuttings appears to be strongly dependent from the type of propagation material as affected by the environment [11] and is therefore considered unsuitable for rapid clonal propagation or crop establishment [3,8]. On the other hand, micropropagation is considered an interesting alternative for rapidly obtaining genetic homogenous and uniform plant material, suitable for more specialized plantings, taking in right consideration the possible somaclonal variations which are often observed in micropropagated plants, probably deriving from the massive use of hormones in the production cycle [12].

In vitro caper propagation was reported for the first time in 1984 [13] and high rooting response was obtained by Rodriguez in 1990 [14]. Afterwards several papers have reported successful micropropagation protocols for *C. spinosa* and other caper related diverse species in different areas [15,16,17,18,19,20,21,22,23,24,25]. Nevertheless, although several authors have reported high success rates (80–100%) for the rooting phase, little attention has been paid to the hypothetic effect of the genotype on the results, as already claimed by Al-Mahmood [22]. Indeed, most of them deal with local unnamed populations or wild individuals, whereas very few reports regard well identified genotypes [23,24] and no one selected cultivated accessions. Furthermore, rooting induction requires high levels of auxins which on the other hand may stimulate callus formation [18,21]. In these cultivation conditions, the abundant presence of callus delays root formation [18] and is unfavorable for ex vitro transfer. Moreover, high concentrations of auxins could be inhibitory for root growth [26] and may be the cause of somaclonal variations in the production cycle [12].

For these reasons, the establishment of a reliable and efficient micropropagation procedure is of particular interest for caper clonal multiplication of selected accessions. In general, the extreme variability of the proposed protocols existing in the literature and of the factors involved, including genotype, together with the erratic response obtained, cause the limitation in the applicability of these protocols, so that optimal combination of factors such as proliferation methods, type, and concentration of PGRs, rooting media, and duration must be better pursued on a case-by-case basis.

Considering the paramount importance in orchard establishment of choosing the best adapted genotypes for a specific area, the objective of the present research work was to develop and optimize micropropagation protocols for in vitro caper rooting of three selected Sicilian accessions, already under commercial cultivation, deemed worthy of further diffusion in the area and in similar cultural environments for their interesting bio-agronomic traits.

## 2. Materials and Methods

Plant material for in vitro rooting trials was obtained from 12-year-old plants of three Sicilian biotypes of *C. spinosa* L.—’Sal 39’, ‘Sal 37’, and ‘Sal 35’—recently selected for their bio-agronomic value, and commercially grown on the Aeolian island of Salina (38°33′49” N). The shoot explants for rooting were obtained following a multiplication phase of nodal segment shoots (1 cm in length) both on a solid substrate or on a liquid substrate in bioreactor (TIS), on a culture medium containing 1.5 mg L^−1^ meta-Topolin + 0.05 mg L^−1^ IBA, as previously described elsewhere [27].

The regenerated shoots of each accession obtained by both the proliferation methods (solid/liquid) were submitted, in separate rooting experiments, to different treatments to compare either the individual effect of different auxins (NAA or IBA or IAA) and concentrations (1 or 2 mg L^−1^) or the effect of an auxin combination (NAA+IBA 0.75 and 0.25 mg L^−1^, respectively). In all the experiments, the MS rooting media were supplemented with different carbon sources (sucrose or fructose) at the same concentration (30 g/L^−1^), as summarized in Table 1. Each treatment included 42 explants. Six replicates (Petri dishes) and seven explants for replicate were used for each trial. The explants were maintained for six days in a climatic growth chamber at 25 ± 1 °C in the dark and then placed under a cool white fluorescent lamp, with a photosynthetic photon flux density (PPFD) of 35 μmol m^−1^ s^−1^ and a photoperiod of 16 h. MS hormone-free rooting media supplemented with 30 g/L^−1^ sucrose or fructose were used as control. After a period of six weeks of rooting phase, data regarding rooting percentage, and root number and length were recorded and submitted to statistical analysis.

In order to verify the possibility of transferring new seedlings from an in vitro to an in vivo environment, an acclimatization protocol has been developed, considering previous experiences [23] and using low environmental impact materials for an easier transplant in a cultivation area.

## 3. Data Analysis

The experimental design was a complete randomization. In order to highlight statistically significant differences and possible interactions between the two factors (cultivar and medium) the two-way analysis of variance (ANOVA) was performed (*p* ≤ 0.05). One-way ANOVA was performed when the interaction between two factors was not significant; each factor was analyzed individually and the separation of the averages was performed by Tukey’s test (*p* ≤ 0.05). When the effects of the different culture media used in the presence of different hormones and carbon sources were expressed as percentages (on a Petri dish basis), the data were arcsin square-root transformed prior to analysis. The statistical analysis was performed using Systat 13 for Windows.

## 4. Results and Discussion

Rooting performance of three *C. spinosa* L. different Sicilian cultivated accessions was evaluated trough the observation of rooting percentage, root number, and length per shoot. In the first experiment, where the effect of the initial proliferation method, auxin type and concentration, and carbohydrate source on rooting was studied, a number of significant differences were noted among the accessions and between the proliferation methods, after six weeks of culture (Table 2 and Table 3). As a whole, explants coming from the PlantForm bioreactors liquid medium supplemented with sucrose (Table 2) showed better rooting performances (up to double rooting percentages) than those which came from the solid medium. When fructose was utilized as an alternative carbohydrate source to sucrose (Table 3) a generalized poor rate of rooting (up to one-third less) was recorded. Among the accessions, ‘Sal 39’ consistently showed better rooting performances, no matter what the treatment was. The ‘Sal 39’ accession gave best results in terms of rooting percentage (57%), root number per shoot (4) and length (1.11 cm) with the explants obtained in the liquid medium (Figure 1 and Figure 2) in the NAA 2 mg L^−1^ treatment supplemented with sucrose (MSuN2) and a minimum of 2%, 0.2 and 0.4, respectively, with IBA 1 mg L^−1^ supplemented with fructose (MFrB1). On the contrary, rooting performances of ‘Sal 37’ and ‘Sal 35’ were consistently unsatisfactory under all the tested conditions. ‘Sal 37’ and ‘Sal 35’ completely failed to root in many of the tested conditions, especially with IBA and IAA, regardless of the previously applied proliferation method and of the PGRs concentration and of the carbon source adopted.

In the second experiment, the effect of an auxin combination (NAA + IBA, 0.75 and 0.25 mg L^−1^, respectively) and the different carbon source (sucrose or fructose, 30 g/L) added to the MS medium was studied (Table 4 and Table 5).

As a whole, explants of all accessions proceeding from the solid medium and cultured in a MS medium supplemented with fructose gave poorer rooting performances than those obtained from PlantForm bioreactors in a MS medium supplemented with sucrose (Table 4 and Table 5). Under these last conditions the highest values of rooting (67%), root number (4.1), and length (0.9 cm) were observed in ‘Sal 39’. Lower values were generally obtained with all the accessions when explants from solid culture were used and when fructose was the carbon source adopted. Overall, these results showed once again a strong genotype effect, being ‘Sal 39’ the most responsive and ‘Sal 35’ the most recalcitrant accession to the applied rooting treatments.

Altogether the results obtained in both the experiments carried out showed that, at the tested conditions, a prevailing effect onto the rooting results was exerted by the genotype. In fact, ‘Sal 39’ microcuttings showed the best rooting performances whatever was the treatment they received, showing to be easier to root than the other accessions. Furthermore, if we compare the results of the auxin-free theses (control = MSuC, MFrC) ‘Sal 39’ rooting performances in both the experiments were generally higher than (or, at the best, similar to) the best results obtained with ‘Sal 37’ and ‘Sal 35’, regardless of the treatment these latter accessions received. Secondly, strong, significant differences were observed as to be related to the type of medium (solid/liquid) adopted in the multiplication phase. Microcuttings proceeding from multiplication in a liquid medium consistently and constantly rooted better than those from the solid medium. Among the tested auxins, NAA, alone or in combination with IBA, was more effective than IBA and IAA. As far as carbon source supplemented to the rooting media is regarded, sucrose generally resulted to be more effective than fructose, confirming what suggested by other authors that sucrose should be the carbohydrate of choice for caper in vitro propagation [25].

Whereas the role and the different effect exerted by auxins in promoting microcuttings in vitro root formation and specific efficient protocols are well documented for a number of woody fruit species, results reported so far in the literature for in vitro caper rooting appear to be affected by an extreme heterogeneity and somewhat contrasting results.

Rodriguez and coworkers [14], comparing the effect of different concentration of IAA, IBA, and NAA on rooting performance of local caper populations of unreported origin, found better rooting results (70%) with the use of IAA at 5,25 mg L^−1^, followed by IBA (50%) and NAA (20%), evidencing strong efficacy differences among auxins and concentrations. They also underlined that population heterogeneity affected proliferation and regeneration results. Ben Salem et al. [17] using 0.5–2.0 mg L^−1^ of IAA, IBA, and NAA in explants coming from one-year old mother plants of *C.spinosa* obtained the maximum rooting (80%) with IAA at 1.5 mg L^−1^. Chalak and Elbitar [18] reported high rooting response (87–92%) of shoots from a selected mature shrub of *C. spinosa* of unspecified origin, after 4 h pulse treatment with IAA 100 mg L^−1^; Caglar et al. [28] improved the rooting percentage of a Lebanese ecotype up to 80.5% with IBA at 5 mg L^−1^ for 10 minutes; Tian et al. [29] obtained rooting percentages up to 87% with IBA at 100–200 mg L^−1^. Musallam et al. [21] reported that the best auxin for in vitro rooting of wild caper plants material was IAA at the level of 5.0 mg L^−1^. Al-Mahmood et al. [22] found for a local unnamed cultivar maximum root formation percentage (60%) with 2.0 mg L^−1^ IAA, but reported similar successful results with IBA and NAA at various concentrations. Carra et al. [23], from unfertilized ovules of two *C. spinosa* selected genotypes, obtained the best results (100% of rooted explants) when explants were dipped for 10 min in 50 µM IBA solution, whereas, in a successive work [24], with a mature plant of a Sicilian caper genotype, they found that IBA, NAA, and IAA stimulated the formation of roots from nodal explants but reported the best result in terms of a high percentage of rooted shoots (93.7%) with the synthetic phenylurea 2,3-MDPU (1 µM). Attia and coworkers [30], using axillary buds of wild *C. spinosa* plants, observed that the highest percentage of the average number of rooted shoots (56.7%) was obtained with 1.5 mg L^−1^ NAA. In another *Capparis* species, such as *C. decidua*, best rooting was achieved on a medium supplemented with IBA (1 mg L^−1^) [16], but in a subsequent work of the same authors [31] 1.0 mg L^−1^IBA combined with 0.5 mg L^−1^ IAA was considered better, whereas Deora and Shekhawat [15] found that 60–70% of the shoots rooted when pulse treated with 100 mg L^−1^ IBA. In another species (*C. orientalis*), closely related to *C. spinosa,* the highest rooting percentage (60%) was obtained with 1 mg L^−1^ of both IBA and NAA [20] and, lastly, Keresa and coworkers [25], working with wild bushes of *C. orientalis*, found that the rooting rate was equally efficient in the media supplemented with 2 mg L^−1^ of IAA or IBA, or in hormone-free rooting medium, whereas NAA at 2 mg L^−1^ was discarded due to the abundant callus formation on the induced roots.

Altogether these results clearly confirm, as already suggested by Carra [23], that for an individual genotype, or even the same local population, the rate of rooting is strongly determined by the type and concentration of auxin, but, considered as a whole, they also seem to indicate that no one protocol may fit all genotypes.

The present study offers an interpretation of the variability of the results available in the literature as to be related to the different rooting attitude showed by different genotypes and confirms previous observations carried out on the same accessions about their different adaptation to in vitro regeneration phase [27].

In our experiments, we found low rooting efficiency of IBA and no response to rooting in IAA treatments, at the tested concentrations. This result is apparently in contrast with most of the results available in the literature for the genus *Capparis* where IAA and IBA are generally considered as the most effective auxins rather than NAA. This contrasting result might be explained by a different sensitivity to PGRs (type and concentration) offered by different genotypes or by carry-over effects from the multiplication phase, or else by a combination of both the factors. Indeed, in vitro rooting is considered genotype-specific within many woody fruit species such as almond [32], pistachio [33], and peach [34], among the others. On the other hand, Keresa et al. [25] evaluated in vitro caper rooting efficiency as affected by the carry-over effect and evoked the role of CKs as auxin antagonists and their involvement in tuning auxin transport and biosynthesis. Reportedly [25], BA (BAP) and its stable derivatives, are suspected to inhibit rooting, whereas meta-Topolin (mT), an aromatic natural cytokinin alternative to BA, and its derivatives do not inhibit in vitro formation, and their breakdown is relatively fast. Nonetheless, they did not find any significant difference in the rooting efficiency between the microshoots previously proliferated in the two different cytokinins, BAP, and mT, probably due to the very low concentration of CKs used during the proliferation phase [25]. In the present study, we only used microcuttings proceeding from microshoots previously proliferated in mT and therefore carry-over effect, if any, should be attributed to the type (liquid or solid) of the medium. It therefore cannot be ruled out that the differences in rooting efficiency we observed in the present study between liquid and solid media could be attributed to the influence exerted by the TIS which proved to promote not only proliferation but also length and vigor of the shoots due to a higher relative growth rate, compared to the solid medium system [27]. Similarly, Gentile and coworkers [35], working with *Prunus* rootstocks, reported for at least one genotype a higher rooting percentage when the culture was proliferated in mT before the rooting induction.

The maximum rooting performances reported here were observed when explants, proceeding from liquid culture in PlantForm and induced in the MS medium with sucrose, as a carbon source, were supplemented with NAA 0.75 mg L^−1^ + IBA 0.25 mg L^−1^, suggesting the importance of the role exerted by the cultural conditions during the proliferation phase. Additionally, these results confirm that mixtures of root-promoting substances are sometimes more effective than either component alone [31,36]. However, these encouraging results were true for only one of the tested accessions (‘Sal 39’), so that an inherent genetic stronger rooting potential can be recognized for this genotype, whereas the other two accessions were revealed to be recalcitrant to the same applied rooting treatments.

## 5. Acclimatization

The in vitro seedlings have been acclimatized by transferring them in conditions of absolute sterility in Jiffy^®^ pots (sterilized in an autoclave for 1 minute) soaked with sterile double-distilled water inside Magenta^®^ GA7 and placed in the growth chamber under the same conditions thermal and light phases of the in vitro phase, until the emission of roots outside the Jiffy^®^ (Figure 3). Subsequently the plantlets with Jiffy^®^ were transferred to plastic containers containing a mixture of peat and perlite (in a ratio of 70/30) and placed under controlled growth conditions. To maintain high relative humidity conditions, plantlets were covered with plastic bags. Relative humidity was therefore reduced gradually and the complete removal of the plastic bag took place after 6 weeks. The plantlets were maintained at 25 ± 1 °C in the dark, and then placed under a cool white fluorescent lamp, with a photosynthetic photon flux density (PPFD) of 35 µmol m^−1^ s^−1^ and a photoperiod of 16 h. They were later placed in a greenhouse for final acclimatization (Figure 4). No negative effects of rapid root browning nor on survival to ex vitro transfer were detected and no particular acclimatization differences emerged, as observed by Carra et al. [23].

## 6. Conclusions

The selection of vigorous and productive plants from wild and cultivated Sicilian caper (*C. spinosa* L.) populations carried out in the recent past by the research activity of the University of Palermo and other institutions gave origin to some selected ‘biotypes’ [7,10]. Three of these Sicilian selections, already spread in commercial orchards in the Aeolian island of Salina, namely ‘Sal 39’, ‘Sal 37’, and ‘Sal 35’, which have proven to be able to produce standard high quality flower buds of excellent firmness, even after processing, and to provide an abundant and constant caper production over the years, with a sufficiently compact harvest period, were considered in the present study, with the aim of developing and optimizing micropropagation protocols for their in vitro propagation.

The carry-over effect of the previous procedure adopted for in vitro multiplication (liquid or solid medium) and, concurrently, the type and concentration of auxin and the carbon source used revealed to be of paramount importance in determining the results of the rooting phase. Nevertheless, it has to be underlined the prevailing significant effect on the induction of *C. spinosa* roots exerted by the genotype of the individual accession. A genotype effect on characteristics such as yield, vegetative vigor, presence of spines, uniformity of flowering, resistance to water stress, caper quality and composition, and conventional rooting is well documented in caper [7,9,10].

This is the first report of a comparative evaluation of the different attitude to in vitro rooting of three commercial caper (*C. spinosa* L) varieties and provides a reliable protocol for in vitro rooting of one of them, namely ‘Sal 39’.

Comparing the results obtained in the experiments carried out in the present work, the highest rooting performances were achieved when the starting microshoots came from liquid culture in the twister and were placed in MS basal medium, sucrose as carbon source, supplemented with a combination of NAA 0.75 mg L^−1^ + IBA 0.25 mg L^−1^. Under these experimental conditions, it was possible to obtain a maximum of 67% of rooted shoots, but only for one of the tested accessions (‘Sal 39’), whereas poorer, unsatisfactory results were obtained with the other two accessions, suggesting a significant different rooting attitude and, thus, a strong genotype effect. This last evidence, unless other new procedures prove their reliability, implies careful consideration in the further diffusion of ‘Sal 37’ and ‘Sal 35’ accessions, at least from the nursery important character of ease of propagation. As a whole, the results of our research, while on one hand may give an explanation to the heterogeneity of the research results obtained so far in the literature, to be related to the genetic heterogeneity of the adopted plant materials, on the other hand strongly indicate the need of further research, focused on an accession-by-accession basis; since it seems unrealistic the search for a unique protocol that fits all genotypes, especially when considering the overall high polymorphism of the genus [37] and the well-known marked heterogeneity existing among the cultivated forms of *Capparis spinosa* L. in Italy [7,38,39].

## Figures and Tables

**Figure 1 plants-09-00398-f001:**
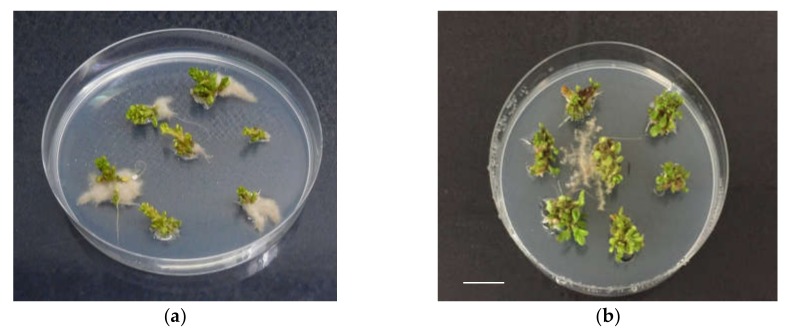
In vitro rooting of shoots of ‘Sal 39’, coming from liquid, in presence of 2 mg L^−1^ of NAA (**a**), (**b**), bar = 1 cm.

**Figure 2 plants-09-00398-f002:**
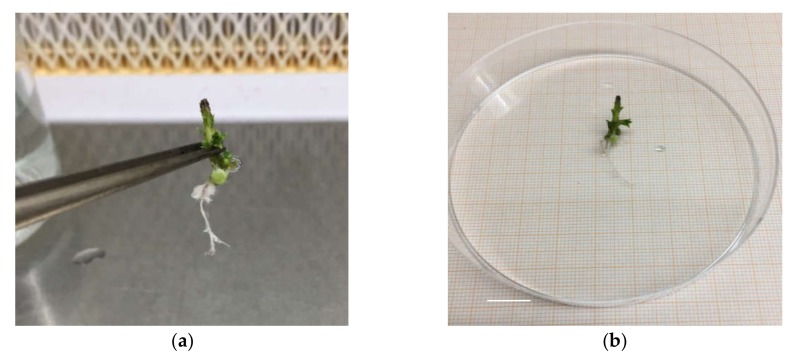
Root explant ‘Sal 39’ (**a**), root length (**b**), bar = 1 cm.

**Figure 3 plants-09-00398-f003:**
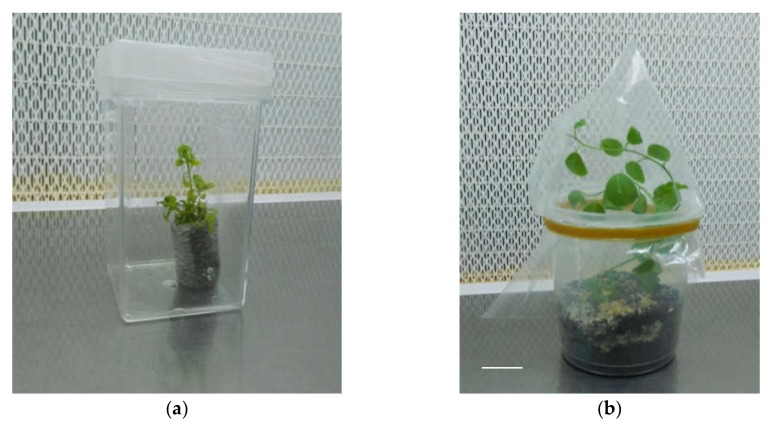
‘Sal 39’ acclimatized in Jiffy^®^ (**a**), in pot (**b**); bar = 1 cm.

**Figure 4 plants-09-00398-f004:**
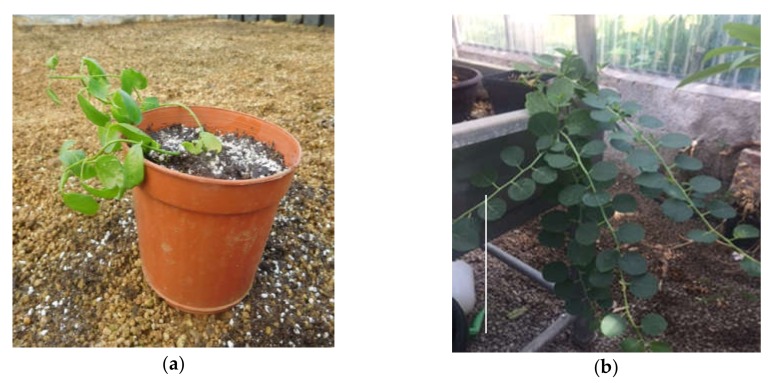
‘Sal 39’ placed in a greenhouse for acclimatization (**a**), (**b**) bar = 10 cm.

**Table 1 plants-09-00398-t001:** Plant growth regulators and carbon sources used in the growth media to induce the formation of adventitious roots from *C. spinosa* shoot explants

Medium	Auxin			Carbon Source	
	NAA mg L^−1^	IAA mg L^−1^	IBA mg L^−1^	Sucrose g/L^−1^	Fructose g/L^−1^
MSuC	0	0	0	30	0
MFrC	0	0	0	0	30
MSuN1	1	0	0	30	0
MSuN2	2	0	0	30	0
MSuA1	0	1	0	30	0
MSuA2	0	2	0	30	0
MSuB1	0	0	1	30	0
MSuB2	0	0	2	30	0
MFrN1	1	0	0	0	30
MFrN2	2	0	0	0	30
MFrA1	0	1	0	0	30
MFrA2	0	2	0	0	30
MFrB1	0	0	1	0	30
MFrB2	0	0	2	0	30
MSuN-B	0,75	0,25	0	30	0
MFrN-B	0,75	0,25	0	0	30

**Abbreviations:** MSu (medium with sucrose); MFr (medium with fructose);N, A, B (medium with NAA or IAA or IBA, respectively); 1, 2 different concentrations (1 or 2 mg L^−1^, respectively); N-B (medium with NAA+IBA, 0.75 and 0.25 mg L^−1^, respectively). MSuC/MFrC (control, hormone-free, medium with sucrose or fructose, respectively).

**Table 2 plants-09-00398-t002:** Conversion rate, mean root number and length, from explants of three accessions of *C. spinosa* L. regenerated from liquid and solid medium, according different auxin and sucrose treatments, after six weeks of culture

Accession	Medium		Liquid			Solid	
		Rooting (%)	Roots Per Shoot (n°)	Root Length (cm)	Rooting (%)	Roots Per Shoot (n°)	Root Length (cm)
‘Sal 39’	MSuC	43 ± 0.5 b	3.0 ± 0.3 ab	0.8 ± 0.1 b	24 ± 0.3 a	1.7 ± 0.2 a	0.64 ± 0.1 a
	MSuN1	36 ± 0.4 b	2.5 ± 0.3 b	0.7 ± 0.1 c	19 ± 0.4 b	1.3 ± 0.2 b	0.58 ± 0.0 b
	MSuN2	57± 0.3 a	4.0 ± 0.2 a	1.1 ± 0.1 a	29 ± 0.2 a	2.0 ± 0.1 a	0.75 ± 0.1 a
	MSuA1	24 ± 0.4 c	1.7 ± 0.2 c	0.6 ± 0.1 d	12 ± 0.2 c	0.8 ± 0.1 c	0.50 ± 0.0 c
	MSuA2	10 ± 0.4 e	0.7 ± 0.1 e	0.4 ± 0.1 e	5 ± 0.5 d	0.3 ± 0.1 c	0.34 ± 0.0 e
	MSuB1	14 ± 0.4 d	1.0 ± 0.1 d	0.6 ± 0.0 d	7 ± 0.3 d	0.5 ± 0.1 c	0.45 ± 0.0 d
	MSuB2	26 ± 0.7 c	1.8 ± 0.3 c	0.7 ± 0.1 c	10 ± 0.2 c	0.7 ± 0.1 c	0.49 ± 0.0 c
‘Sal 37’	MSuC	12 ± 0.3 b	0.8 ± 0.1 b	0.6 ± 0.0 c	7 ± 0.3 b	0.5 ± 0.1 b	0.37 ± 0.0 b
	MSuN1	10 ± 0.4 b	0.7 ± 0.1 c	0.7 ± 0.1 b	0	0	0
	MSuN2	17 ± 0.4 a	1.2 ± 0.2 a	0.9 ± 0.1 a	12 ± 0.3 a	0.8 ± 0.1 a	0.67 ± 0.05 a
	MSuA1	0	0	0	0	0	0
	MSuA2	0	0	0	0	0	0
	MSuB1	0	0	0	0	0	0
	MSuB2	0	0	0	0	0	0
‘Sal 35’	MSuC	5 ± 0.4 b	0.3 ± 0.1 b	0.1 ± 0.0 c	2 ± 0.4 b	0.17 ± 0.1 b	0.24 ± 0.03 b
	MSuN1	2 ± 0.4 b	0.2 ± 0.1 c	0.4 ± 0.0 b	0	0	0
	MSuN2	7 ± 0.5 a	0.5 ± 0.1 a	0.6 ± 0.1 a	5 ± 0.4 a	0.3 ± 0.1 a	0.43 ± 0.05 a
	MSuA1	0	0	0	0	0	0
	MSuA2	0	0	0	0	0	0
	MSuB1	0	0	0	0	0	0
	MSuB2	0	0	0	0	0	0

Data were collected after six weeks from the beginning of the experiment and each treatment comprises 42 explants. The different letters grouped for each single accession indicate statistically significant differences; n.s. not significant (Tukey’s test, *p* ≤ 0.05).

**Table 3 plants-09-00398-t003:** Conversion rate, mean root number and length, from explants of three accessions of *C. spinosa* L. regenerated from liquid and solid medium, according different auxin and sucrose treatments, after six weeks of culture

Accession	Medium	Liquid			Solid		
		Rooting (%)	Rootsper Shoot (n°)	Rootlength (cm)	Rooting (%)	Roots Per Shoot (n°)	Root Length (cm)
‘Sal 39’	MFrC	21 ± 0.6 b	2.2 ± 0.2 b	0.86 ± 0.07 a	17 ± 0.3 b	1.5 ± 0.2 a	0.42 ± 0.05 a
	MFrN1	24 ± 0.7 b	2.7 ± 0.4 a	0.80 ± 0.05 b	19 ± 0.3 b	1.2 ± 0.1 b	0.39 ± 0.05 b
	MFrN2	38 ± 0.6 a	3 ± 0.3 a	0.87 ± 0.05 a	29 ± 0.2 a	1.8 ± 0.2 a	0.44 ± 0.05 a
	MFrA1	10 ± 0.8 d	0.7 ± 0.3 d	0.46 ± 0.06 d	12 ± 0.3c	0.5 ± 0.1 c	0.24 ± 0.05 d
	MFrA2	14 ± 0.5 c	1 ± 0.2 c	0.44 ± 0.05 d	17 ± 0.3 b	0.7 ± 0.1 c	0.34 ± 0.06 c
	MFrB1	2 ± 0.4 e	0.2 ± 0.1 e	0.40 ± 0.04 e	5 ± 0.3 d	0.2 ± 0.1 d	0.11 ± 0.04 e
	MFrB2	5 ± 0.4 e	0.3 ± 0.1 e	0.58 ± 0.08 c	7 ± 0.3 d	0.5 ± 0.1 c	0.29 ± 0.04 d
‘Sal 37’	MFrC	2 ± 0.4 d	1 ± 0.2 b	0.52 ± 0.05 a	14 ± 0.5 b	0.8 ± 0.1 c	0.35 ± 0.05 a
	MFrN1	10 ± 0.4 b	0.7 ± 0.1 c	0.40 ± 0.06 b	7 ± 0.3 d	0.3 ± 0.1 e	0.28 ± 0.04 b
	MFrN2	26 ± 0.8 a	1.8 ± 0.3 a	0.54 ± 0.04 a	19 ± 0.5 a	1.5 ± 0.2 a	0.38 ± 0.04 a
	MFrA1	12 ± 0.5 b	0.8 ± 0.1 c	0.39 ± 0.03 c	5 ± 0.4 d	0.5 ± 0.2 d	0.29 ± 0.03 b
	MFrA2	7 ± 0.5 c	1.5 ± 0.2 a	0.43 ± 0.05 b	12 ± 0.4 c	1 ± 0.2 b	0.20 ± 0.04 c
	MFrB1	2 ± 0.4 d	0.3 ± 0.1 d	0.36 ± 0.03 c	0	0	0
	MFrB2	0	0	0	0	0	0
‘Sal 35’	MFrC	0	0	0	0	0	0
	MFrN1	0	0	0	0	0	0
	MFrN2	0	0	0	0	0	0
	MFrA1	0	0	0	0	0	0
	MFrA2	0	0	0	0	0	0
	MFrB1	0	0	0	0	0	0
	MFrB2	0	0	0	0	0	0

Data were collected after six weeks from the beginning of the experiment and each treatment comprises 42 explants. The different letters grouped for each single accession indicate statistically significant differences; n.s. not significant (Tukey’s test, *p* ≤ 0.05).

**Table 4 plants-09-00398-t004:** Conversion rate, mean root number and length, from explants of three accessions of *C. spinosa* L. regenerated from liquid and solid medium, on MSuC and MSuN-B medium

Accession	Medium	Liquid			Solid		
		Rooting (%)	Roots Per Shoot (n°)	Root Length (cm)	Rooting (%)	Roots Per Shoot (n°)	Root Length (cm)
‘Sal 39’	MSuC	36 ± 0.3 ns	2.3 ± 0.2 ns	0.5 ± 0.06 ns	12 ± 0.1 ns	1.4 ± 0.4 ns	0.30 ± 0.1 ns
	MSuN-B	67 ± 0.3 *	4.1 ± 0.2 *	0.9 ± 0.07*	17 ± 0.2 *	2.7 ± 0.2 *	0.70 ± 0.08 *
‘Sal 37’	MSuC	14 ± 0.1 ns	1 ± 0.3 ns	0.4 ± 0.07 ns	7 ± 0.1 ns	0.6 ± 0.2 ns	0.40 ± 0.07 ns
	MSuN-B	19 ± 0.2 *	1.9 ± 0.3 *	0.8 ± 0.08*	10 ± 0.1 *	1.1 ± 0.3 *	0.60 ± 0.08 *
‘Sal 35’	MSuC	10 ± 0.1 ns	0.1 ± 0.05 ns	0.1 ± 0.04 ns	5 ± 0.1 ns	0.20 ± 0.1 ns	0.10 ± 0.04 ns
	MSuN-B	14 ± 0.2 *	0.3 ± 0.2 *	0.4 ± 0.08 *	12 ± 0.2 *	0.20 ± 0.1 ns	0.20 ± 0.05*

The data show means ± s.e.; with 42 explants in each treatment; ns, * show respectively non-significant and significant differences for each accession according to the medium used at *p* ≤ 0.05.

**Table 5 plants-09-00398-t005:** Conversion rate, mean root number and length, from explants of three accessions of *C. spinosa* L. regenerated from liquid and solid medium, on MFrC and MFrN-B medium

Accession	Medium	Liquid			Solid		
		Rooting (%)	Roots Per Shoot (no.)	Root Length (cm)	Rooting (%)	Roots Per Shoot (no.)	Root Length (cm)
‘Sal 39’	MFrC	33 ± 0.5 *	2.6 ± 0.6 ns	0.68 ± 0.05 ns	7 ± 0.5 ns	0.9 ± 0.2 ns	0.24 ± 0.03ns
	MFrN-B	21 ± 0.5 ns	3.2 ± 0.5 *	0.77 ± 0.06 *	14 ± 0.4 *	1.4 ± 0.3 *	0.35 ± 0.06 *
‘Sal 37’	MFrC	7 ± 0.5 *	1.3 ± 0.3 ns	0.57 ± 0.02 *	2 ± 0.4 ns	0.5 ± 0.3 ns	0.21 ± 0.04ns
	MFrN-B	5 ± 0.4 ns	1.5 ± 0.3 *	0.47 ± 0.06 ns	10 ± 0.4 *	0.7 ± 0.2 *	0.27 ± 0.04 *
‘Sal 35’	MFrC	5 ± 0.4 ns	0.7 ± 0.2 ns	0.21 ± 0.04 ns	0	0	0
	MFrN-B	0	0	0	0	0	0

The data show means ± s.e.; with 42 explants in each treatment; ns, * show respectively non-significant and significant differences for each accession according to the medium used at *p* ≤ 0.05.

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
