# Peer review of "In Vitro Rooting of Capparis spinosa L. as Affected by Genotype and by the Proliferation Method Adopted During the Multiplication Phase"

_plants, 2020, doi:10.3390/plants9030398_

Round 1

Reviewer 1 Report

The research described in the manuscript are very interesting and of high importance for being the first research focused on the study of the impact of different genotypes of the same plant (Capparis spinose L.) to the efficacy of their multiplication and in vitro rooting.

Until now, different micropropagation protocols were studied on the level of the species within the genus Capparis, but not on the level of the genotypes within the species, and especially not on the level of selected cultivated genotypes (i.e. selected biotypes) which are oh high producing and economic value. The studies until now were focused mainly on the local populations and wild individual plants of caper.

Until now, the studies emphasize the problem of caper rooting (difficult-to-root woody species) in the conventional way, and various micropropagation protocols for rapid clonal propagation or crop establishment were studied. Taking into account that in some Mediterranean areas, such as Sicily, selected cultivated accessions of caper are cultivated, representing a high-revenue crop, the results shown in this paper will significantly contribute to the overall technological advancement of the caper production.

The authors clearly detected and explained the problem, and set up the experiment as complete randomization in multiple repetitions with controls. They used appropriate statistical tools and clearly interpreted the obtained results. The authors draw clear conclusions on the basis of the interpreted results.

The text is very clear and understandable, guiding the reader through the whole research, beginning with the problem definition, the way of experimentation, the interpretation of the results, until the conclusions.

The paper, as an original scientific work, will have strong scientific impact and it will be interesting for scientific community.

However, the clarification is needed for the following:

  1. Sal 35, Sal 37, and Sal 39 - what kind of plant material exactly is this about? Are these material cultivars, biotypes or accessions, or something else? From the description of the material and their economic value, I conclude this material could be cultivars. It this material are accessions obtained from the gene bank, in that case they could be referred to as accessions, regardless if they are cultivars, populations, breeding lines, wild capers, etc. However, you have not emphasize that these material was obtained from the gene bank (with accession ID). If your material are cultivars, please emphasize this, and write their names in single quotation marks, eg. ‘Sal 39’.

Explanation: you use four terms for the same material: accessions, selected cultivated genotypes, Sicilian biotypes, Sicilian selection, and cultivars. In the Introduction, you use the term selected cultivated genotypes. In the Materials and methods, you use the term biotypes in the part where you describe the plant material, but later in the same chapter under the description of statistical analysis (eg. two-way variance analysis), one of your factors is cultivar. More further in the manuscript, eg. in Tab 2. and within the text, you use the term accession. In the Conclusion chapter, you use the term Sicilian selection.

There are also a need for some minor corrections:

  1. Missing a blank space after the full stop of the sentence, and before the next sentence. (Please, see under lines no. 5, 57, 78, 88, 128, 193)
  2. Missing a blank space between two words (Lines no. 33, 55, 58, 126, 195, 197, 198, 199, 203, 207, 209, and 211).
  3. In vitro should be written in italics. Please, correct in the whole manuscript.
  4. mg/L should be corrected to mg L-1 (Lines no. 81, 82, 84, 89, 126, 128, 146, 191, 193, 195, 197, 199, 200, 201, 209, 211, 213, 214, 216, 217, 252, and 312; Table 1. and in abbreviations under Tab. 1.)
  5. In the Materials and methods, the amount of sucrose and fructose is referred to as mg/L, and in the Tab. 1 you are referring to it as g/L. Please, adjust the units.
  6. g/L correct to g L-1 (in Table 1., and line no. 147)
  7. „playa“ correct to „play a“ (line no. 38)
  8. Within the References:
    1. Ref No. 1. – Please, indicate the source where the paper or book chapter is published, or this is some other type of publication?
    2. Ref No. 2, 6, 7, 8, 11, 14-24, 27, 28, and 39. – Missing a blank space between the journal title and year of the issue.
    3. Ref No. 3. –when referring the authors, you need the semicolon mark (;) - see the ref. No. 2 for example.
    4. Ref No. 5. – Elalcaparo correct to El alcaparro, and 19177 correct to 19/77

Author Response

Dear Reviewer,

Thanks for all the comments which definitively helped us to improve the paper. In the attached file you can find all the answer to your comments step by step.

Thanks again

Regards

The Authors

Reviewer 2 Report

The manuscript ´In vitro rooting of Capparis spinosa L. as affected by genotype and by the proliferation method adopted during the multiplication phase ’, provides interesting and useful information on different methods for micropropagation of this species.

• The Introduction, Material and Methods, Results and Discussion are clear and well written.
• The methodology is appropriate to achieve the specific aims, but in some parts, the authors have to give more information. (see comments in the manuscript)
• Improve figure and table captions.
• Additional comments and suggestion of corrections are directly reported on the manuscript.

The manuscript cannot be accepted for publication in the present form, but it needs minor revisions.

Author Response

(The authors gave the same response as above.)
